# Effectiveness of Psychological Treatments for Borderline Personality Disorder and Predictors of Treatment Outcomes: A Multivariate Multilevel Meta-Analysis of Data from All Design Types

**DOI:** 10.3390/jcm10235622

**Published:** 2021-11-29

**Authors:** Sophie A. Rameckers, Rogier E. J. Verhoef, Raoul P. P. P. Grasman, Wouter R. Cox, Arnold A. P. van Emmerik, Izabella M. Engelmoer, Arnoud Arntz

**Affiliations:** 1Department of Clinical Psychology, University of Amsterdam, 1018 WS Amsterdam, The Netherlands; w.r.cox@uva.nl (W.R.C.); a.a.p.vanemmerik@uva.nl (A.A.P.v.E.); a.r.arntz@uva.nl (A.A.); 2Department of Developmental Psychology, University of Utrecht, 3584 CS Utrecht, The Netherlands; r.e.j.verhoef@uu.nl; 3Department of Psychological Methods, University of Amsterdam, 1018 WS Amsterdam, The Netherlands; r.p.p.p.grasman@uva.nl; 4Institute for Personality Disorders and Behavioral Problems, de Viersprong Amsterdam, 1115 HG Duivendrecht, The Netherlands; izabella.engelmoer@deviersprong.nl

**Keywords:** Borderline Personality Disorder, psychotherapy, meta-analysis, treatment effectiveness, multilevel

## Abstract

We examined the effectiveness of psychotherapies for adult Borderline Personality Disorder (BPD) in a multilevel meta-analysis, including all trial types (PROSPERO ID: CRD42020111351). We tested several predictors, including trial- and outcome type (continuous or dichotomous), setting, BPD symptom domain and mean age. We included 87 studies (N = 5881) from searches between 2013 and 2019 in four databases. We controlled for differing treatment lengths and a logarithmic relationship between treatment duration and effectiveness. Sensitivity analyses were conducted by excluding outliers and by prioritizing total scale scores when both subscale and total scores were reported. Schema Therapy, Mentalization-Based Treatment and reduced Dialectical Behavior Therapy were associated with higher effect sizes than average, and treatment-as-usual with lower effect sizes. General severity and affective instability showed the strongest improvement, dissociation, anger, impulsivity and suicidality/self-injury the least. Treatment effectiveness decreased as the age of participants increased. Dichotomous outcomes were associated to larger effects, and analyses based on last observation carried forward to smaller effects. Compared to the average, the highest reductions were found for certain specialized psychotherapies. All BPD domains improved, though not equally. These findings have a high generalizability. However, causal conclusions cannot be drawn, although the design type did not influence the results.

## 1. Introduction

The Diagnostic and Statistical Manual of Mental Disorders-5 (DSM-5) Borderline Personality Disorder (BPD) diagnosis is based on nine criteria, such as feelings of emptiness, affective instability, suicidality and difficulties controlling anger [1]. The estimated prevalence is 1.1% in the Netherlands [2,3], 2.7% in the United States and 0.7% in Great Britain [4]. In addition, comorbidity with other disorders is high [5], and a staggering 75% of BPD patients attempt suicide at least once in their life, and 10% of patients actually commit suicide [6].

BPD was historically viewed as a difficult-to-treat or even untreatable disorder [7], but in the last 30 years this view has drastically and positively changed. Moreover, although BPD treatments are associated with high dropout rates, this might not be as high as previously assumed [8]. The present meta-analysis will summarize the treatment outcomes of psychological treatments (not pharmacological treatments) for BPD.

There are many non-specialized psychological treatments available, but four psychological treatments have been specifically developed for BPD (i.e., the Big-4): Dialectical Behavior Therapy (DBT), Schema Therapy (ST), Transference Focused Psychotherapy (TFP) and Mentalization-Based Treatment (MBT). Unfortunately, it is difficult to gain more insight into the relative effectiveness of these treatments because of several reasons. First, existing mutual comparisons of these so-called Big-4 psychotherapies for BPD often are inadequately powered and are, therefore, far from conclusive, e.g., [9,10]. Second, as so many treatments are available, it is virtually impossible to mutually compare all existing treatments to each other in well powered randomized controlled trials (RCTs), let alone to replicate such trials. The lack of RCTs has forced previous meta-analyses to investigate BPD treatments (specialized and non-specialized) together as a group, finding superiority of specialized treatments compared to control conditions (*g* = 0.32) [11] and superiority of BPD treatments compared to treatment-as-usual (TAU; *d* = 0.59) [12] in reducing BPD severity. This is in line with findings of a recent meta-analysis [13], which found that psychotherapies compared to TAU were superior in reducing general BPD severity (*d* = 0.52), self-harm (*d* = 0.32) and suicide-related outcomes (*d* = 0.34). Direct comparisons between specific specialized treatments and TAU suggested the superiority of DBT in reducing BPD severity (*d* = 0.60) and self-harm (*d* = 0.28), and the superiority of MBT compared to TAU in reducing self-harm (RR = 0.62) and suicidality (RR = 0.10). Although this meta-analysis demonstrates that it is possible to compare at least some treatments, the evidence for these effects remains limited due to the low number of direct comparisons between treatments [13]. As a result, it is difficult for patients, therapists, policy makers and researchers to estimate the efficacy of specific BPD treatments based on traditional meta-analyses of direct comparisons.

In addition, there are many aspects to these treatments and many study characteristics that can also play an important role in determining effectiveness. However, the sheer number of these aspects makes it impossible to study each in combination with all other factors in RCTs. These important treatment factors are treatment format [14,15,16] and setting [17], but also treatment duration. There is a lot of heterogeneity in treatment duration, and even though it was suggested that treatment duration is not related to treatment outcomes [11], it is an important factor to include because there might be a dose–response relationship. Furthermore, patient characteristics such as comorbidity, age, gender and substance abuse [9,18,19,20,21,22] could be important. Additionally, interesting study characteristics are trial type and study quality [23] and the handling of missing data. Lastly, publication year might be interesting to study, as one would expect that the effectiveness of treatments has increased given the radical change over the last decades in how BPD is viewed and in the development of new treatment models.

The meta-analytic approach that maximizes internal validity is examining the between-treatment effects of RCTs [24]. However, while continuing to pursue experimental comparisons of BPD treatments and predictors of treatment outcomes, different meta-analytic approaches may also shed light on this issue and help broaden our scope. Instead of meta-analyzing the effect sizes representing the difference between two study arms in an RCT, we therefore meta-analyzed the effect sizes representing the change on BPD indices within study arms. This approach offers several advantages. First, it still allows comparisons between treatment approaches and other predictors across studies. In fact, it even allows for comparisons between treatments and certain predictors that cannot be made based on traditional meta-analyses. While traditional meta-analyses are restricted by the number of studies that are available for certain comparisons, this approach can estimate effects based on a specified model and will thus be able to include more data. Second, by employing a multilevel meta-analytic approach, we can include multiple indices of BPD-pathology. Third, and perhaps most important, it allows the inclusion of the large number of uncontrolled studies that characterize the BPD treatment literature. This offers the opportunity to statistically summarize the many studies available, while not being restricted by a specific study design. Effectiveness studies of other designs are often conducted in clinical routine mental healthcare settings, an important point also raised by other authors [23]. Therefore, while the inclusion of non-RCT designs will decrease internal validity, this approach enhances external validity and could offer additional relevant insights that can be applied in clinical practice.

In sum, the present meta-analysis aims to summarize BPD treatment outcome studies in a new way. By including multiple trial and study designs, it will be able to make use of a much larger part of the BPD treatment literature. Examining study characteristics such as trial type will further allow us to estimate their threats to validity. We will address the following questions related to the effectiveness of psychological treatments for BPD, in which we will combine the data from all interventions and outcomes:

(1) Do psychological treatments differ in effectiveness?; (2) Do BPD criteria differ in the degree to which they change during psychological treatment?; (3) Are there specific treatments that are especially effective for particular BPD criteria?; (4) Is treatment format (i.e., individual, group, combined) related to treatment effectiveness?; (5) Is treatment setting (i.e., inpatient, day-treatment, outpatient) related to treatment effectiveness?; (6) Is there a dose–effect relationship between length of treatment and reduction in BPD manifestations?; (7) Did treatments become more effective during the last decades?; (8) Are demographic characteristics (mean age, gender composition, comorbidity with axis I and axis II disorders, exclusion of substance use disorders, mean educational level) related to treatment effectiveness?; (9) Are study-specific characteristics (i.e., study quality and type, missing data handling, medication prescribed, country of testing, outcome type, assessment type) related to treatment effectiveness?

## 2. Materials and Methods

### 2.1. Identification and Selection of Studies

The present meta-analysis followed the PRISMA Guidelines [25] and was preregistered at PROSPERO (ID: CRD42020111351). The PRISMA-P table is shown in Appendix A. A database search was performed and updated in Pubmed, Embase, Cochrane and Psychinfo (search dates: 21 June 2013, 4 February 2015, 6 March 2019 and 29 October 2019). Reference lists of reviews, meta-analyses and other manuscripts were also checked. One submitted master thesis was obtained. Appendix A provides the search terms and results per database of the last literature update. Different combinations of eight raters (AvE, AA, RV, SR, MM, WC, IE, PC) independently selected studies for inclusion. In the case of disagreement, a decision was reached through further discussion.

Inclusion criteria were applied to all study arms separately. First, we only included studies of psychological BPD treatments with the following design types: RCT, open trial, case series or cohort study. Second, we only included studies with adult patients (age ≥ 18) with a primary diagnosis of BPD according to the DSM-III, DSM-III-R or DSM-IV(-Tr) criteria. If an English abstract was available and passed the initial screening, language was not an eligibility criterion.

We also defined several exclusion criteria. First, studies of treatments of ‘double diagnoses’ (i.e., testing treatments developed for a specific combination of two diagnoses, such as BPD and eating disorder) were excluded, because such treatments are adapted to the combination of diagnoses, and their effects cannot be generalized to BPD treatments focused on treating BPD specifically. Second, for the same reasons, we excluded mixed personality disorders samples, unless separate statistics on the BPD subsample were reported and the treatment was specifically directed at BPD. A tolerance of 10% was allowed (i.e., at least 90% had to meet a full BPD diagnosis). Third, we excluded single case studies (unless consecutive with *n* ≥ 5) because there is little guarantee that reporting is not biased (i.e., only reported in case of success). Fourth, we excluded treatments consisting of subsets of techniques or modules which are clearly not intended to be complete treatments. However, we did include tests of reduced protocols intended to be complete treatments (e.g., ‘reduced DBT’ for DBT treatments missing a specific ingredient). Fifth, we excluded treatment modules that are explicit additions to treatments and not intended to be complete treatments, such as STEPPS, specific courses or skills training. Sixth, we excluded forensic samples. These samples require specific forms of treatment, and effects must be evaluated in a forensic context and are difficult to generalize to more common BPD samples, which was an important aim of our present meta-analytic approach. Finally, we excluded studies that did not report post-treatment effectiveness data on BPD-pathology as defined by the DSM-criteria, or if we could not transform the data into effect sizes. We contacted authors in case information was missing.

### 2.2. Treatment Classification

We initially classified treatments into 17 categories. A ‘mixed’ category was created for treatments that combined elements of two or more treatments. We then reduced the number of treatment categories by collapsing categories with *n* < 100 and the mixed category into a ‘specified others’ category. We classified DBT treatments as full DBT if they included the four standard DBT components and as reduced DBT if not all four components were present. Control conditions that received some form of standard active care were labeled as TAU. As Community Treatment by Experts (CTBE) constitutes a more stringent comparison condition than TAU, e.g., [26], we distinguished it as a separate category.

### 2.3. Coding of Methodological Quality

We assessed separate treatment arms of included studies for risk of bias by evaluating nine design criteria, adapted from [24] to fit the focus of the present meta-analysis (see Appendix A). We calculated the mean quality score based on all items that were applicable to the study arm. Following calibration on a subset of included studies, different pairs of six coders (AvE, RV, SR, MM, CvE) completed the checklist for the remaining studies. The interrater agreement based on intra-class correlations (ICCs) per item ranged from 0.841 to 0.980, with an average of 0.895 (median = 0.888). The internal consistency of the mean score was 0.691, with an ICC of 0.948.

### 2.4. Coding of Other Characteristics

Three raters (KM, WC and SR) independently coded the other sample, treatment and study characteristics. If necessary, authors were contacted by email to clarify issues. If studies reported total scale scores in addition to separate subscale scores (e.g., BPDSI overall and BPDSI specific criteria), we only included the subscale scores in the primary analysis. For some predictors, we reduced the number of categories. This was the case for countries which we reduced to three country groups (reduced to: Europe, USA, Australia/Canada/New Zealand), trial type (reduced to: RCT, non-RCT, open trial), missing data handling (reduced to: modern Intent-to-Treat (ITT) techniques such as multilevel analysis, Last-Observation-Carried-Forward (LOCF, Unclear ITT techniques, Completer analysis), setting (reduced to: Inpatient, Day Treatment, Outpatient), format (reduced to: Individual, Group, Combined), and substance use exclusion (reduced to: No exclusion, Exclusion if needing clinical detox, Exclusion of dependence and abuse). We coded outcomes as dichotomous if changes were expressed in proportions (number of patients meeting a criterion), and as continuous if effect sizes were based on means and standard deviations. When studies had mixed consecutive treatment settings (e.g., first inpatient, then outpatient) the first setting was coded. This was decided as day treatment or inpatient treatment always preceded outpatient treatment, and it was assumed that most treatment effects would be observed during the first phase, which had a higher intensity. Two raters (IE and ASV) coded a randomly selected subset of studies and disagreements were resolved through discussion. The average ICC of the coded statistics and demographic variables was 0.991 (0.926–1.000).

### 2.5. Effect Size Definition

We expressed within-treatment changes per outcome variable in terms of Cohen’s d, with the (pooled per study) baseline standard deviation (SD) as denominator and the mean change as numerator. These effect sizes have been proven to be the most robust, precise and least biased [27]. For dichotomous outcomes, we defined Cohen’s d according to the standard formula after applying the Agresti–Coull correction to observed proportions (see Appendix A), as this yields more valid estimates and can deal with instances where 0% or 100% is reported [26]. Lastly, all effect size estimates were transformed to Hedges’ g (see Appendix A).

Included treatments greatly varied in length, ranging from six weeks to over two years. We therefore estimated the relationship between treatment duration in weeks and Hedges’ g, which appeared to be logarithmic, *F* (1, 563) = 724.34, $R^{2}$ = 0.563, *p* < 0.001. To compare treatment lengths, treatment effects need to be estimated at a fixed number of weeks. Therefore, we transformed Hedges’ g (and accompanying standard errors) to account for the logarithmic relationship between treatment duration and outcomes, and to represent the effect size at one year of treatment (i.e., transformed effect sizes; see Appendix A). We also controlled for this non-linear relationship by adding the log transformed number of weeks as a covariate to each model (i.e., untransformed effect sizes) and calculated the marginal means of all categorical predictors at one year of treatment (52 weeks). 

### 2.6. Statistical Analysis

To control for dependency in effect sizes [28] we conducted a three-level meta-analysis to analyze the effect sizes of the pre-post changes during treatment using the Metafor package [29] in R. Missing values in predictors were handled by imputing 20 sets, using predictive mean modeling and including all available predictors.

We used multimodel inference from the MuMin package [30] as a model selection procedure. We chose the highest-ranked model based on the Bayesian Information Criterion (BIC). When models had a BIC difference between zero and two, we treated them as equivalent [31] and selected the model with the significantly highest log-likelihood value. Two predictors were forced into each model: treatment and BPD outcome domain.

The main effects from the final models were investigated with deviation contrasts on the estimated marginal means, which were estimated based on all possible combinations of predictor levels. For each categorical predictor, we compared the effect size for each level of that predictor to the average effect (i.e., grand mean) of all levels of that predictor. For example, differences between interventions or BPD outcome domains were examined by comparing each individual intervention or outcome domain to the average effect of all interventions or outcome domains. We examined the treatment by domain interaction (research question 3) by subsetting our dataset for each domain. We controlled for multiple testing by applying the Holm correction [32].

We calculated Egger’s test by adding the standard error as an additional covariate to the model and inspected the structure of the data with funnel plots of the residuals. To our knowledge, there is no automated trim-and-fill procedure available for multilevel models [29], and it is unclear how the trim-and-fill procedure performs in three-level models [33]. Therefore, we explored the output of the trim-and-fill procedure on a model without moderators and by treating all effect sizes as independent, to assess possible patterns of bias in the data. We identified outliers with studentized residuals (absolute value > 3). In addition, we conducted several sensitivity analyses. We reran the analyses without outliers, but also on the total severity scores instead of subscale scores.

## 3. Results

### 3.1. Search Results

The final selection included in the analyses consisted of 110 records concerning a total of 87 studies (N = 5881) (see Figure 1). The interrater agreement, Fleiss’ kappa [34], between different sets of three raters ranged from 0.553 to 1.000, with an average of 0.778. The percentage agreement ranged from 87.10% to 100%. Appendix A displays all included studies and their characteristics, Appendix A their references. A complete table with all outcomes and effect sizes was published online (https://osf.io/htxmq, accessed on 23 November 2021).

We included 35 RCTs, 46 open trials and 6 non-RCT trials. The mean publication year was 2010 (1991–2019). The studies were most frequently conducted in the USA (*n* = 30), Germany (*n* = 12), the Netherlands (*n* = 11) and the UK (*n* = 10). The average proportion of males was 0.14 (0.00–0.56). The mean age of the sample was 30.99 (20.50–40.10). The final selection consisted of 11 treatment categories. There were 33 treatment arms for DBT (*n* = 2503), 11 for ST (*n* = 263), 10 for DBTmin (*n* = 273), 9 for MBT (*n* = 468), 9 for Psychodynamic Treatment (PDT, *n* = 488), 6 for Cognitive Behavior Therapy (CBT, *n* = 161), 5 for TFP (*n* = 157), 24 for TAU (*n* = 724), 2 for CTBE (*n* = 101) and 4 for mixed therapies (*n* = 363). Several treatments were collapsed into a specified others category (*n* = 700), consisting of Cognitive Analytic Therapy, Interpersonal Therapy, Client-Centered Therapy, Structural Clinical Management, General Psychiatric Management, Therapeutic Community and Dynamic Deconstructive Psychotherapy. The average number of weeks of therapy was 46.57 (median = 52), ranging from 6 to 156 weeks.

Three variables contained missing data: the mean number of Axis I and II disorders and educational (ISCED) level. However, these variables were excluded, as adding the imputed variables to our models did not improve the fit (Dm (3, 198.39) = 0.03, *p* = 0.993).

### 3.2. Main Analysis

#### 3.2.1. Main Analysis Model Selection

The main analysis was run on 87 studies and 535 effect sizes. The within-study variance ($\sigma_{u}^{2}$ = 0.142, $X^{2}$(1) = 555.490, *p* < 0.001) and the between-study variance ($\sigma_{u}^{2}$ = 0.138, $X^{2}$(1) = 77.181, *p* < 0.001) were significant. The percentage of sampling variance was 13.42%, the within-study variance was 44.02% and the between-study variance was 42.56%. The upper part of Table 1 shows the results for the full model.

After we fitted all possible models, there were two equivalent models (ΔBIC < 2). The first model consisted of three predictors (BPD domain, treatment and outcome type), whereas the second model consisted of four predictors: the predictors of the first model plus mean age. The second model had the highest significant log-likelihood ratio test statistic, so we chose this model as our final model. The tests for the predictors in the final model are shown in the lower part of Table 1. For the final model, the within-study variance ($\sigma_{u}^{2}$ = 0.085, $X^{2}$(1) = 215.176, *p* < 0.001) and the between-study variance ($\sigma_{u}^{2}$ = 0.078, $X^{2}$(1) = 44.718, *p* < 0.001) were significant. The percentage of sampling variance was 20.98%, the within-study variance was 41.23% and the between-study variance was 37.79%. In our final model, age was negatively related to treatment outcomes. In other words, as the mean age of the sample increased, the effect of treatment decreased.

#### 3.2.2. Deviation Contrasts

We examined the deviation contrasts for each of the selected categorical predictors in the final models. Each level of the predictors was compared to the average estimated mean’s effect, which was large (*g* = 0.921). All effect sizes were included in the analyses.

Treatment. Overall, all treatments appeared effective in reducing overall BPD symptomatology, with effect sizes ranging from medium to large. At one year, ST (*g* = 1.233) followed by reduced DBT (*g* = 1.123) was associated with larger effect sizes in reducing BPD symptoms compared to the grand mean of all interventions. Compared to this average, TAU (*g* = 0.588) was associated with smaller effect sizes (see Table 2 and Figure 2).

BPD outcomes. At one year, the largest improvement compared to the average improvement over all domains was found for general severity (*g* = 1.317), followed by affective instability (*g* = 1.267) (see Figure 3). The least improvement was found for dissociation (*g* = 0.662), impulsivity (*g* = 0.703), anger (*g* = 0.707) and suicidality (*g* = 0.722).

Outcome type. Continuous outcomes (*g* = 0.752) were associated with smaller effect sizes, whereas dichotomous outcomes (*g* = 1.089) were related to larger effect sizes.

#### 3.2.3. Treatment and BPD Interaction

We explored the interaction between treatment and BPD outcome domain (see Table 3). The grand means ranged between 0.422 and 1.384. At one year, TAU was associated with smaller improvements in improving general severity (*g* = 0.641), impulsivity (*g* = 0.320), suicidality (*g* = 0.393), emptiness (*g* = 0.311) and anger (*g* = 0.234). Compared to the average treatment effects, reduced DBT was related to larger improvements in anger (*g* = 0.829) and affective instability (*g* = 2.569). ST (*g* = 1.161) and MBT (*g* = 0.872) were related to larger improvements in suicidality.

#### 3.2.4. Outliers and Bias

There were 29 outliers in the final model (5.42%), which were effect sizes ranging from 0.277 to 4.174. These outliers came from seven different treatments: ST, reduced DBT, DBT, Specified-other treatment studies, TAU, CBT and MBT. Of the outliers, 20 came from RCT designs and 25 were based on continuous outcomes. All effect sizes from the Reiss et al., (2014–1) study were labeled as outliers. These effect sizes were all larger than 3.189, which was unrealistically large compared to the other effect sizes in our sample. The same applied to the one outlying variable from Farrell et al., (2019). Two outliers from Jacob et al., (2018) and Nordahl et al., (2005) were relatively small: 0.277 and 0.560, respectively.

Egger’s test was significant, indicating a relationship between stronger effects and lower precision, *F* (1, 512) = 369.820, *p* < 0.001, *B* = 5.764. The funnel plot also indicated the presence of outliers and did not show a symmetric pattern of the residuals (see Appendix A). According to the trim-and-fill procedure on the overall model, 28 effect sizes were missing on the right side of the funnel plot. No effect sizes were missing on the left side of the funnel plot. Simulating these additional studies would increase the average effect size to *g* = 0.047.

When all outliers were removed, Egger’s test was still significant, *F* (1, 483) = 146.027, *p* < 0.001, *B* = 3.481. The funnel plot of the final model without outliers is shown in Appendix A. In the final model, all predictors were significant. The estimated marginal means showed several differences. The grand estimated mean was 0.827. In addition to ST and reduced DBT, MBT was also associated with larger effect sizes compared to the average treatment effect. CTBE was, in addition to TAU, associated with smaller effect sizes. The only other observed difference was that identity disturbance also showed stronger improvement compared to the average treatment effect. For outcome type, no differences were observed (see Appendix A).

### 3.3. Sensitivity Analysis

#### 3.3.1. Sensitivity Analysis Model Selection

The second sensitivity analysis, conducted on 447 effect sizes, prioritized total scale results if included studies reported these in addition to subscale results. Model fit did not improve when we added the three imputed variables with missing data, Dm (3, 135.81) = 0.02, *p* = 0.997.

Both the within-study variance ($\sigma_{u}^{2}$ = 0.112, $X^{2}$(1) = 317.769, *p* < 0.001) and the between-study variance ($\sigma_{u}^{2}$ = 0.185, $X^{2}$(1) = 100.139, *p* < 0.001) were significant. The percentage of sampling variance was 13.73%, the within-study variance was 32.61% and the between-study variance was 53.67%. The upper part of Appendix A shows the results for the full model.

After our model selection procedure, there were two equivalent models. The first model consisted of treatment, BPD domain and outcome type. The second model also included mean age. Based on comparisons of the log-likelihood value, the model with treatment, BPD domain, outcome type and mean age was selected. The tests for the final model are shown in the lower part of Appendix A. The within-study variance ($\sigma_{u}^{2}$ = 0.052, $X^{2}$(1) = 91.958, *p* < 0.001) and the between-study variance ($\sigma_{u}^{2}$ = 0.102) $X^{2}$(1) = 65.485, *p* < 0.001) of the final model were also significant. The percentage of sampling variance was 23.54%, the within-study variance was 25.67% and the between-study variance was 50.79%.

#### 3.3.2. Sensitivity Analysis Deviation Contrasts

For this sensitivity analysis, we conducted the same analyses on the predictors selected in the final models. The average grand mean effect was large (*g* = 0.875).

Treatment. At one year, ST (*g* = 1.273) followed by reduced DBT (*g* = 1.060) were associated with higher effect sizes compared to the average treatment effect, whereas TAU (*g* = 0.545) and CTBE (*g* = 0.670) were associated with smaller effect sizes (see Appendix A).

BPD outcome. At one year, general severity (*g* = 1.337) and affective instability (*g* = 1.119) showed the strongest change compared to the average grand mean effect of all effect sizes. Impulsivity (*g* = 0.575), anger (*g* = 0.648), dissociation (*g* = 0.677) and suicidality (*g* = 0.716) were associated with the lowest effect sizes compared to the average (see Appendix A).

Outcome type. Similar to the other analyses, continuous outcomes (*g* = 0.723) were related to weaker outcomes, whereas dichotomous outcomes (*g* = 1.027) were related to stronger outcomes.

#### 3.3.3. Sensitivity Analysis Outliers and Bias

Again, Egger’s test was significant, *F* (1, 424) = 142.833, *p* < 0.001, *B* = 3.403. The trim-and-fill procedure indicated that 17 residuals were missing on the right, whereas none were missing on the left side. Simulating these studies would increase the average effect size with 0.036. There were 23 outliers (5.4%). Ten outliers were from general severity, and eleven from suicidality.

In the final model without outliers, all predictors were significant, as was Egger’s test, *F* (1, 401) = 62.165, *p* < 0.001, *B* = 1.866. Both funnel plots are shown in Appendix A. The grand mean of all predictors was 0.815. Again, estimated means decreased. The only difference was that reduced DBT (*g* = 0.922) was no longer associated with higher effect sizes, but that MBT (*g* = 0.953) was significant instead. No further differences for BPD domain and outcome type, compared to the analysis with outliers included, were observed (see Appendix A). 

The sensitivity analyses on the untransformed effect sizes are reported in Appendix A). Any differences between both methods are reported in the discussion section.

## 4. Discussion

This meta-analysis addressed several questions regarding the BPD treatment literature in a new way. To estimate treatment effects across BPD domains, we included the pre- to post-treatment data from all treatments and outcome domains.

The effect sizes estimated at one year indicated that ST and reduced DBT showed the strongest changes in BPD outcome domains compared to the average treatment effect. Based on the outlier analyses, MBT was also related to larger effect sizes compared to the average. The untransformed sensitivity analyses indicated larger effect sizes of ST and MBT, and, less robustly, also of CBT. TAU and, to a lesser degree, CTBE were related to weaker improvements in all BPD outcome domains. The findings for CTBE were not as robust as those for TAU, which is not surprising as CTBE is generally viewed as an optimized variant of TAU [35]. The difference of TAU compared with the average effect was small, but medium compared to ST and reduced DBT. Together, these findings suggest that mainly the specialized treatments, i.e., ST, MBT and reduced DBT, appear to yield the largest effect sizes in the treatment of BPD compared to the average of all treatments. Overall, it should be emphasized that the average effectiveness of the treatments was moderate to large. With regard to the individual treatment effects, each treatment was compared to the average effect of all treatments. Thus, if some treatments were found to be related to larger effect sizes, this does not imply any specific differences between two individual treatments (as this is a different comparison that requires a more specific, more direct test).

Compared to the average, reduced DBT, and not DBT, was related to higher treatment effects. This is surprising, because it suggests that, compared to all treatments, a reduced version of DBT, and not the complete treatment model, is related to larger effect sizes, although the effect size difference between both treatments was small (*g* = 0.226). This is in accordance with one previous meta-analysis that showed that DBT yielded a moderate effect size compared to TAU, but a small effect size compared to specialized treatments [23]. However, a few factors should be considered when interpreting this finding. First, the effective ingredients of reduced DBT have not yet been identified. Second, reduced DBT treatment models were very heterogeneous (i.e., studies omitted different DBT components), thereby possibly influencing these findings. Third, it is possible that some elements of DBT do not contribute to its effectiveness.

We also examined the changes in separate BPD domains. All domains showed moderate to large improvements, but the strongest improvement was observed for general BPD severity and affective instability. Impulsivity, suicidality/self-injury, anger and dissociation showed the least improvement compared to the average. Complex BPD symptoms such as suicidality/self-injury, impulsivity and intense anger have indeed been identified as relatively resistant to change [36]. In contrast with our findings, one study showed that affective instability was relatively resistant to change during a two-year treatment [37]. However, these findings cannot easily be compared to ours as it was unclear if and what type of treatment patients received in this study. Interestingly, a recent network study found that affective instability was not only central to the BPD criteria but was also central to the changes in these criteria [38]. This means that affective instability is strongly related to other symptoms and that changes in affective instability are an important mechanism for change in other symptoms. This partly supports the finding that the strongest changes compared to the average were observed for this symptom. An alternative reason why some domains might show less improvement is that specific treatments may focus more strongly on some symptom domains than on others.

Our exploration of the interaction between treatment and BPD criteria was largely consistent with our main findings. With large effect sizes, ST and MBT were robustly related to larger reductions in suicidality compared to the average treatment effect. Reduced DBT showed the largest improvements in anger and affective instability. In the untransformed sensitivity analysis, ST was related to the largest change for the criteria that showed the least improvement overall (i.e., impulsivity, suicidality, anger and dissociation), while DBT was associated with strong improvements in anger. Based on all analyses, with effect sizes ranging from small to large, TAU was related to smaller outcomes for seven BPD criteria, i.e., general severity, impulsivity, suicidality, emptiness, anger, affective instability and dissociation. These findings are in line with the general trends of our study, as the treatments with the highest (and lowest) effect sizes also had similar effect sizes on separate outcome domains. This is a very relevant issue for clinical practice, as more knowledge about these interactions can improve the development and effectiveness of personalized treatment in which patients are matched to treatments based on their symptom profiles. Thus, these findings raise interesting new leads for future studies, but more research is necessary to further test and examine these interactions.

Although the present study used a different approach compared to more traditional meta-analyses, our findings are consistent with earlier studies [11,12] and with the most recent meta-analysis in this area [13]. In most cases, these previous studies have not found any differences between the specialized treatments and other protocolized treatments, but only with TAU. The effect sizes in the meta-analysis by Storebø et al., (2020) and the present one, representing differences between TAU and all other treatments in reducing general BPD severity (*d* = −0.52 vs. *g* = −0.74) and suicide-related outcomes (*d* = −0.34 vs. *g* = −0.25), were very similar. Moreover, where Storebø et al., (2020) found an effect size of *d* = −0.60 comparing DBT and TAU in reducing BPD severity, the present study found a difference of *g* = −0.76, and our findings also suggested that TAU was less effective compared to the joint other treatments in reducing general severity. Similarly, Storebø et al., (2020) also found that MBT, compared to TAU, was more effective in reducing suicidality. There were, however, differences in the number of studies that were available for each of these treatments. While many DBT trials were included, the number of ST, MBT and reduced DBT studies was smaller (between 9 and 11 studies). Therefore, it is possible that the meta-analytic findings of the specialized treatments with fewer studies were more prone to sources of bias. This is an important reason why it is important to also conduct more primary treatment studies focusing on these specialized models, as was also noted in previous meta-analyses, e.g., [11,13].

Our models also identified several additional predictors of BPD treatment outcomes. Higher age was related to smaller improvements in BPD severity, consistent with the idea that personality becomes more resistant to change with increasing age [22,39]. Even though this effect appeared to be small, the difference in treatment effectiveness can become quite large when the difference in age increases. To illustrate, there is a medium difference in effect between a 20-year-old patient and a 40-year-old patient of 0.42. However, we should interpret this finding with caution as, within individual studies and treatment arms and due to Simpson’s paradox [40,41] in which aggregation of effects can lead to a reversal of effects, the opposite could also be true.

In addition, the largest improvement was found on outcomes describing the number of patients who still meet the DSM criteria for BPD after treatment, compared to outcomes based on continuous scores. Dichotomous outcomes (i.e., proportions) are measured on a different scale and are usually based on some cut-off score. Consequently, patients might still fulfill some criteria to a rather severe degree and suffer from BPD manifestations that do not qualify for a BPD diagnosis. Continuous outcome measures might therefore be less sensitive to this bias. A final interesting predictor was the way trials handled missing data. Completer analyses are generally not recommended because they use biased samples [42]. Therefore, the fact that we did not observe differences between completer analyses and modern ITT techniques was unexpected. Interestingly, for the untransformed effect sizes, LOFC, with a moderate effect size, was related to a smaller effect compared to other analysis types. This is in line with the often-heard suggestion that LOFC methods are very conservative and thus produce smaller effect sizes [43].

Our findings also suggest that treatment format, setting, education level, assessment type, substance use exclusion, medication policy, country of testing and male proportion are not related to treatment outcomes. However, the proportion of males was generally very low and could also be indicative of a gender bias. The true proportion of male patients with BPD is likely higher [3] than in the meta-analyzed studies, which decreases the generalizability of our findings to male BPD patients. Moreover, as publication year was not related to treatment outcomes, there are no indications that, despite all developments, psychological treatments of BPD have improved over the years. Note, however, that the most effective specialized treatments were developed and tested rather recently (the mean publication years for MBT and ST were 2014 and 2012, respectively). In addition, earlier trials might have been conducted on a smaller-scale and might have been more susceptible to bias. Larger and more recent trials could have had a better methodological quality. Therefore, it is possible that while treatments improved, these effects were cancelled out by an improvement in overall study quality. However, we found no relationship between trial type and study quality, and the treatment outcomes, which corresponds to earlier findings [23], supporting our decision to include non-RTCs and uncontrolled studies. Also, we found indications that longer BPD treatments were more effective, but this is inconsistent with earlier findings [11]. Lastly, our findings were relatively robust to outliers and appeared unaffected by publication bias [44].

Several questions remain unanswered by the present meta-analysis. First, the interaction between treatment type and BPD domains was not included in our final models, since several BPD domains were underrepresented or not addressed by (several) treatments at all. We strongly recommend that future treatment studies investigate all BPD-criteria to better document the effectiveness profiles of specific treatments. Second, future meta-analyses should include follow-up treatment effects, because some BPD domains might need more time to improve, and treatments might differ in their long-term effectiveness. Third, it is important to examine the dose–response relationship of treatment effects in more detail, given the observed relationship of time with treatment effectiveness and the finding that inpatient and day-treatments were not related to higher treatment outcomes compared to outpatient treatment. Fourth, future studies should examine the effectiveness of the separate treatment ingredients (such as in DBT), because this is relevant from a cost and time efficiency standpoint. Fifth, the findings of the present study are based on adults with BPD diagnosed according to the DSM (a categorical system), and therefore it is unclear to what degree these findings generalize to younger populations with BPD or diagnoses based on dimensional systems, such as the International Classification of Diseases, 11th edition [ICD-11] [45]. However, the ICD-11 offers a borderline pattern qualifier, which has criteria that are based on the DSM-5 BPD criteria [42]. Thus, it is not expected that using the ICD-11 as a diagnostic system would lead to a highly dissimilar patient group or have a strong influence on the generalizability of the findings. Sixth, we did not examine the cost-effectiveness of these treatments. Such studies are relatively scarce, and more attention should be focused on conducting cost-effectiveness analyses. Studies do suggest that specialized treatments, although more expensive, are cost-effective [46,47].

There are several limitations to our study, and we are aware that all conclusions should be interpreted with these limitations taken into consideration. First, while the inclusion of non-RCT designs greatly enhances the generalizability of the findings, it limits the possibility to draw causal conclusions. Our main aim was therefore to provide an extensive summary of the present BPD treatment outcome literature and to identify overall patterns that should then be further tested in future primary studies. However, our findings are similar to findings of other, more traditional meta-analyses [11,12,13], and trial type (e.g., RCT or non-RCT) was not related to treatment outcomes. Second, while the present study included many predictors, we might have overlooked other important predictors. Third, some treatment categories (e.g., TAU, reduced DBT) were quite heterogenous. While this is difficult to avoid in a meta-analysis of such a large and broad sample of studies, it reveals a lack of consistency in the included studies that also impacts other meta-analyses. The field would therefore greatly benefit if research would focus on fewer treatments and a single set of outcomes (and measures) to facilitate comparisons between studies. Including such a broad range of studies might also have caused heterogeneity in other factors, such as specific characteristics of the included samples. Fourth, we were forced to reduce some predictors to a smaller number of categories that, as a result, became more heterogenous and more difficult to interpret. Fifth, to control for dependency in effect sizes, we analyzed the data in a multilevel random effects model. Although several authors recommend this method [48,49], its validity has not yet been fully established. Several methods have been proposed to handle dependency between effect sizes [28], such as aggregating data, e.g., [11], or using the correlation between instruments. However, such methods introduce new difficulties or lead to a loss of power and valuable data [48]. Thus, adopting a multilevel method was viewed as the best method for our purposes. Lastly, the observed log-linear relationship between treatment duration and outcomes is in accordance with the dose–effect relationships found in other studies [50,51]. However, it should be noted that we modelled the time–effect relationship, and that each model is imperfect. We therefore also used a second method to correct for differences in treatment length by adding treatment duration as a covariate. While imperfect, it is nevertheless an important improvement over other meta-analyses that generally neglect this problem. Note that even when only RCTs are included, the fact that differences between treatment and control arms might change over time is ignored by collapsing RCTs with different time windows.

In conclusion, psychotherapy in general was more effective compared to TAU. Moreover, three of the Big-4 treatments, ST, MBT and reduced DBT, were associated with relatively larger effect sizes compared to the average of all treatments. Second, our findings suggest that treatments should adopt a stronger focus on the BPD symptom domains that show relatively small improvements. These domains, such as dissociation, suicidality and anger, proved to be more resistant to change and improved at a slower rate. However, the fact that these domains still improve over the course of treatment is an encouraging finding. Third, longer treatments were related to larger effect sizes and early intervention seems preferable, as age was negatively related to treatment effectiveness. A positive finding is that many variables, such as gender and substance use exclusion, were not related to treatment outcomes. Fourth, the lack of effects from treatment setting suggests that limited resources are better spent on good outpatient psychotherapy than on relatively expensive inpatient and day treatments. However, there is a strong need for more RCTs that test direct differences between the Big-4 treatments. Taken together, we believe that this summary of the current state of affairs helped to shed light on some uncertainties in the field and generated some interesting new ideas to study in future research.

## Figures and Tables

**Figure 1 jcm-10-05622-f001:**
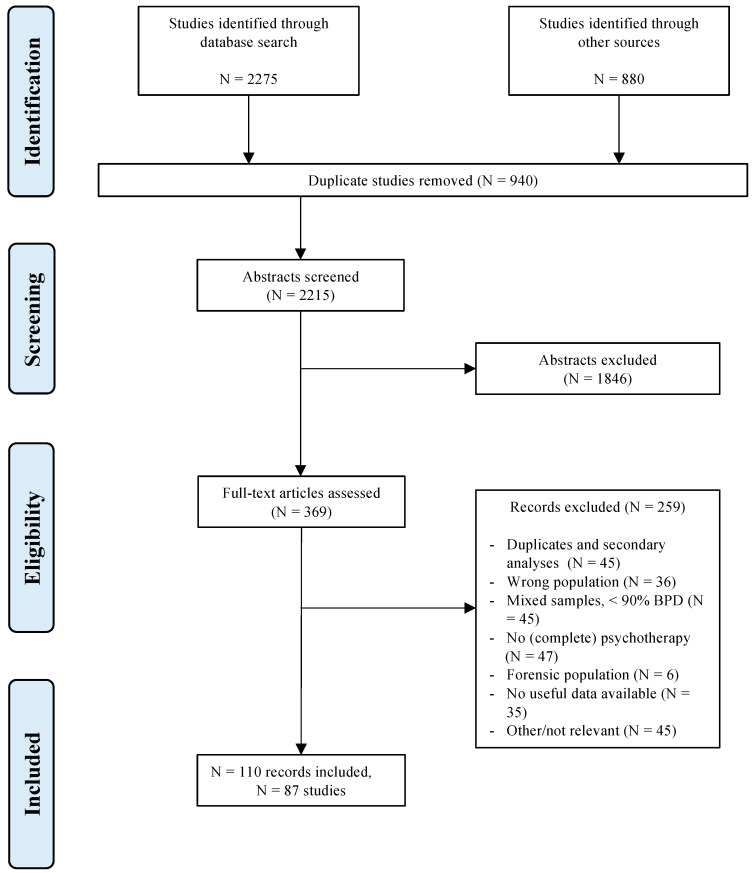
Flowchart of the study selection process.

**Figure 2 jcm-10-05622-f002:**
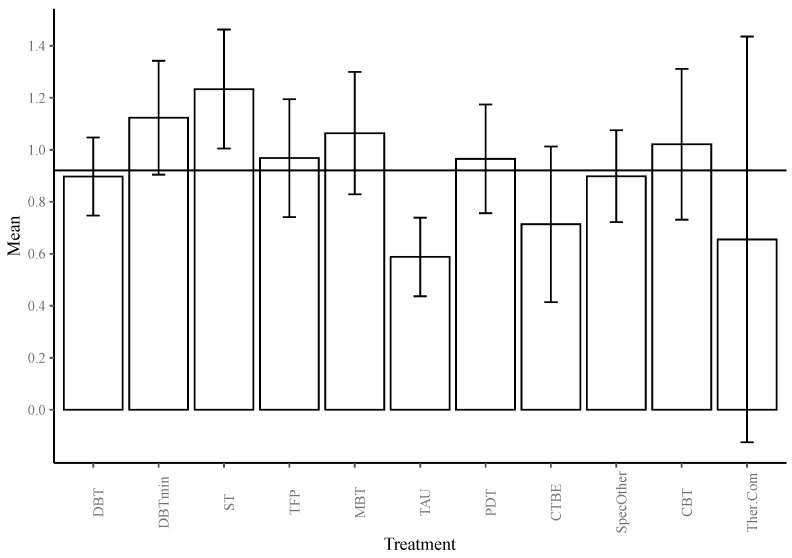
Bar graph of the estimated marginal means of the transformed effect sizes based on the log of time, estimated at one year of treatment for all treatments, with 95% CIs. The grand mean is depicted as a horizontal line at *g* = 0.921.

**Figure 3 jcm-10-05622-f003:**
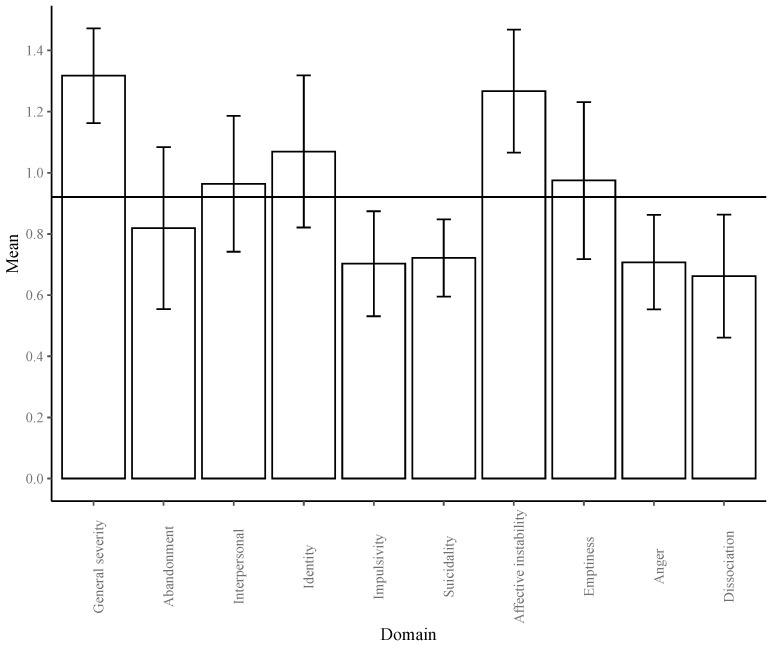
Bar graph of the estimated marginal means of the transformed effect sizes based on the log of time, estimated at one year of treatment for all BPD domains, with 95% CIs. The grand mean is depicted as a horizontal line at *g* = 0.921.

**Table 1 jcm-10-05622-t001:** Chi Square Tests of all the Predictors in the Full and Final Model of the Transformed Effect Sizes.

Variable	$X^{2}$	*df*	Log-Likelihood	*p*
**Full Model**		106	−281.410	
Treatment	156.326	34	−359.572	<0.001 ***
BPD Domain	248.573	35	−405.696	<0.001 ***
Treatment * BPD domain	116.105	44	−339.462	<0.001 ***
Setting	2.504	104	−282.662	0.286
Format	2.944	104	−282.882	0.229
Quality	0.021	105	−281.420	0.885
Trial type	1.213	104	−282.016	0.545
Publication year	0.005	105	−281.412	0.942
Country of testing	1.020	104	−281.919	0.601
Male proportion	0.041	105	−281.430	0.839
Analysis type	20.043	103	−291.431	<0.001 ***
Mean age	11.553	105	−287.186	0.001 **
Medication policy	0.358	105	−281.588	0.550
Substance use exclusion	1.185	104	−282.002	0.548
Assessment type	2.100	103	−282.653	0.553
Outcome type	8.902	105	−285.860	0.003 **
**Final Model**		24	−353.108	
Treatment	69.982	14	−388.099	<0.001 ***
BPD Domain	132.254	15	−419.235	<0.001 ***
Outcome type	11.757	23	−358.986	0.001 **
Mean age	5.413	23	−355.814	0.020 *

Note. * *p* < 0.05. ** *p* < 0.01. *** *p* < 0.001.

**Table 2 jcm-10-05622-t002:** Results of All Deviation Contrasts of the Transformed Effect Sizes.

Contrast	Mean	95% CI	∆*g*	∆*g* (*se*)	∆*g* (*t*)	∆*g* (*p*)	∆*g* (*p’*)
Average Age	*(B)* −0.021	[−0.039, −0.003]		(*se*) 0.009	(*t*) −2.333	0.020 *	
**Treatment**							
DBT	0.897	[0.747, 1.047]	−0.024	0.062	−0.381	0.352	1.000
DBTmin	1.123	[0.904, 1.342]	0.203	0.098	2.063	0.020 *	0.178
ST	1.233	[1.005, 1.462]	0.313	0.104	3.013	0.001 **	0.014 *
TFP	0.968	[0.741, 1.194]	0.047	0.096	0.492	0.311	1.000
MBT	1.064	[0.829, 1.299]	0.143	0.108	1.328	0.092	0.646
TAU	0.588	[0.437, 0.739]	−0.333	0.064	−5.165	<0.001 ***	<0.001 ***
PDT	0.965	[0.756, 1.174]	0.045	0.091	0.490	0.312	1.000
CTBE	0.714	[0.414, 1.013]	−0.207	0.132	−1.571	0.058	0.467
Spec. Other	0.898	[0.722, 1.075]	−0.022	0.079	−0.279	0.390	1.000
CBT	1.021	[0.731, 1.311]	0.100	0.136	0.737	0.231	1.000
Th. Com	0.655	[−0.125, 1.435]	−0.265	0.368	−0.722	0.235	1.000
*Grand Mean*	*0.921*						
**BPD Domain**							
General severity	1.317	[1.162, 1.472]	0.397	0.057	6.966	<0.001 ***	<0.001 ***
Abandonment	0.819	[0.554, 1.084]	−0.102	0.106	−0.960	0.169	0.506
Interpersonal	0.964	[0.742, 1.186]	0.044	0.085	0.517	0.303	0.596
Identity Disturbance	1.069	[0.821, 1.318]	0.149	0.099	1.509	0.066	0.264
Impulsivity	0.703	[0.531, 0.874]	−0.218	0.060	−3.620	<0.001 ***	0.001 **
Suicidality/Self-injury	0.722	[0.595, 0.848]	−0.199	0.046	−4.336	<0.001 ***	<0.001 ***
Affective Instability	1.267	[1.066, 1.468]	0.346	0.074	4.659	<0.001 ***	<0.001 ***
Emptiness	0.975	[0.718, 1.231]	0.054	0.102	0.530	0.298	0.596
Anger	0.707	[0.553, 0.862]	−0.213	0.051	−4.211	<0.001 ***	<0.001 ***
Dissociation	0.662	[0.461, 0.863]	−0.258	0.076	−3.411	<0.001 ***	0.002 **
*Grand Mean*	*0.921*						
**Outcome Type**							
Continuous	0.752	[0.631, 0.873]	−0.123	0.050	−2.451	0.007**	0.015*
Dichotomous	1.089	[0.885, 1.293]	0.123	0.050	2.451	0.007**	0.015*
*Grand Mean*	*0.920*						

Notes. * *p* < 0.05. ** *p* < 0.01. *** *p* < 0.001.

**Table 3 jcm-10-05622-t003:** Deviation Contrasts of the Interaction between Treatment and BPD Domain at One Year.

Domain	Mean (*g*)	95% CI	∆*g*	∆*g* (*se*)	∆*g* (*t*)	∆*g* (*p*)	∆*g* (*p’*)
**General Severity (*n* = *80*)**							
DBT	1.400	[1.028, 1.772]	0.016	0.179	0.087	0.466	1.000
DBTmin	1.689	[1.109, 2.269]	0.304	0.264	1.154	0.126	0.883
ST	1.331	[0.923, 1.739]	−0.053	0.197	−0.269	0.394	1.000
MBT	1.361	[0.936, 1.785]	−0.024	0.199	−0.119	0.453	1.000
TAU	0.641	[0.335, 0.946]	−0.743	0.148	−5.035	<0.001 ***	0.001 **
PDT	1.627	[1.129, 2.125]	0.243	0.230	1.058	0.147	0.883
Spec. Other	1.482	[1.070, 1.895]	0.098	0.195	0.502	0.309	1.000
CBT	1.544	[0.896, 2.192]	0.160	0.293	0.544	0.294	1.000
*Grand Mean*	*1.384*						
**Abandonment (*n* = *8*)**							
ST	0.689	[0.355, 1.024]	0.234	0.128	1.832	0.063	0.190
TAU	0.466	[−0.035, 0.967]	0.011	0.153	0.071	0.473	0.473
Spec. Other	0.210	[−0.316, 0.735]	−0.245	0.157	−1.561	0.090	0.190
*Grand Mean*	*0.455*						
**Interpersonal (*n* = *17*)**							
DBT	0.789	[−0.217, 1.794]	−0.060	0.416	−0.144	0.444	1.000
ST	0.742	[0.075, 1.409]	−0.106	0.318	−0.334	0.372	1.000
TAU	0.645	[−0.011, 1.301]	−0.203	0.315	−0.645	0.265	1.000
Spec. Other	0.946	[0.081, 1.810]	0.097	0.373	0.260	0.400	1.000
CBT	1.121	[−0.524, 2.766]	0.272	0.622	0.438	0.335	1.000
*Grand Mean*	*0.849*						
**Identity (*n* = *14*)**							
ST	1.157	[0.533, 1.782]	0.587	0.289	2.032	0.035 *	0.139
TAU	0.299	[−0.300, 0.897]	−0.272	0.255	−1.064	0.156	0.468
Spec. Other	0.648	[−0.037, 1.334]	0.078	0.284	0.274	0.395	0.523
CBT	0.177	[−1.521, 1.876]	−0.393	0.593	−0.663	0.261	0.523
*Grand Mean*	*0.570*						
**Impulsivity (*n* = *50*)**							
DBT	0.610	[0.367, 0.853]	0.010	0.102	0.096	0.462	1.000
DBTmin	0.851	[0.451, 1.251]	0.251	0.184	1.364	0.090	0.629
ST	0.587	[0.238, 0.937]	−0.013	0.157	−0.080	0.468	1.000
TFP	0.487	[0.149, 0.825]	−0.113	0.138	−0.816	0.210	1.000
TAU	0.320	[0.098, 0.542]	−0.280	0.104	−2.691	0.005 **	0.041 *
PDT	0.597	[0.222, 0.973]	−0.003	0.156	−0.017	0.493	1.000
Spec. Other	0.736	[0.421, 1.052]	0.136	0.138	0.988	0.164	0.986
CBT	0.611	[0.079, 1.143]	0.011	0.235	0.047	0.481	1.000
*Grand Mean*	*0.600*						
**Suicidality (*n* = *184*)**							
DBT	0.563	[0.424, 0.702]	−0.083	0.079	1.054	0.147	1.000
DBTmin	0.711	[0.466, 0.957]	0.065	0.124	0.526	0.300	1.000
ST	1.161	[0.669, 1.653]	0.515	0.229	2.253	0.013 *	0.128
TFP	0.522	[0.199, 0.845]	−0.124	0.144	−0.858	0.196	1.000
MBT	0.872	[0.612, 1.132]	0.226	0.130	1.733	0.042 *	0.358
TAU	0.393	[0.225, 0.560]	−0.253	0.090	−2.826	0.003 **	0.029 *
PDT	0.512	[0.128, 0.897]	−0.134	0.183	−0.730	0.233	1.000
CTBE	0.380	[0.046, 0.713]	−0.266	0.151	−1.763	0.040	0.358
Spec. Other	0.593	[0.390, 0.797]	−0.053	0.104	−0.508	0.306	1.000
CBT	0.765	[0.428, 1.103]	0.119	0.164	0.729	0.234	1.000
Th. Com	0.633	[−0.180, 1.446]	−0.013	0.379	−0.035	0.487	1.000
*Grand Mean*	*0.646*						
**Affective Instability (*n* = *27*)**							
DBT	1.070	[0.491, 1.649]	−0.234	0.291	−0.806	0.215	0.860
DBTmin	2.569	[1.562, 3.576]	1.265	0.433	2.919	0.004 **	0.030 *
ST	1.324	[0.648, 2.000]	0.019	0.306	0.064	0.475	1.000
TFP	0.920	[0.030, 1.811]	−0.384	0.389	−0.988	0.167	0.837
TAU	0.705	[0.187, 1.223]	−0.599	0.249	−2.402	.013 *	0.078
Spec. Other	1.209	[0.563, 1.856]	−0.095	0.305	−0.311	0.379	1.000
CBT	1.333	[−0.446, 3.131]	0.028	0.754	0.038	0.485	1.000
*Grand Mean*	*1.304*						
**Emptiness (*n* = *9*)**							
ST	1.118	[0.519, 1.718]	0.431	0.218	1.975	0.048 *	0.096
TAU	0.311	[−0.257, 0.880]	−0.376	0.166	−2.264	0.032 *	0.096
Spec. Other	0.631	[−0.065, 1.328]	−0.056	0.191	−0.291	0.390	0.390
*Grand Mean*	*0.687*						
**Anger (*n* = *102*)**							
DBT	0.418	[0.294, 0.541]	−0.116	0.068	−1.704	0.046	0.229
DBTmin	0.829	[0.569, 1.088]	0.295	0.121	2.434	0.008 **	0.059
ST	0.679	[0.372, 0.986]	0.146	0.141	1.035	0.152	0.303
TFP	0.626	[0.356, 0.897]	0.093	0.125	0.747	0.228	0.303
TAU	0.234	[0.073, 0.394]	−0.300	0.081	−3.677	<0.001 ***	0.002 **
PDT	0.400	[0.216, 0.585]	−0.133	0.090	−1.469	0.073	0.238
CTBE	0.306	[−0.010, 0.622]	−0.227	0.144	−1.573	0.060	0.238
Spec. Other	0.773	[0.490, 1.057]	0.240	0.132	1.826	0.035 *	0.213
*Grand Mean*	*0.533*						
**Dissociation (*n* = *23*)**							
DBT	0.411	[0.224, 0.599]	−0.011	0.085	−0.133	0.448	0.737
ST	0.472	[0.216, 0.728]	0.049	0.105	0.470	0.322	0.737
TAU	0.316	[0.039, 0.592]	−0.107	0.107	−1.001	0.165	0.659
Spec. Other	0.491	[0.251, 0.731]	0.069	0.098	0.702	0.246	0.737
*Grand Mean*	*0.422*						

Notes. * *p* < 0.05. ** *p* < 0.01. *** *p* < 0.001.

## Data Availability

A complete table with all outcomes and effect sizes was published online (https://osf.io/htxmq, accessed on 23 November 2021).

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
