# Peer review of "Effectiveness of Psychological Treatments for Borderline Personality Disorder and Predictors of Treatment Outcomes: A Multivariate Multilevel Meta-Analysis of Data from All Design Types"

_jcm, 2021, doi:10.3390/jcm10235622_

Round 1

Reviewer 1 Report

I have no comments. The paper is well written and the method seems sound. From my view, this is very relevant research. 

- For which age range is the review valid? As BPD can be treated in youth, I would be interested in this point as an addition to the current paper.
- Does the change in diagnostic system for personality disorders (dimensional approach for diagnosing personality disorder in the ICD-11) affect the results of the paper? Could the authors comment on this point?
- Can the authors further comment on treatment costs/efficiency? Many low resource settings will need efficient interventions. 
- The authors should review the paper to see if any passages can be made shorter or omitted by focussing on the key messages of the paper.   

Reviewer 2 Report

Thanks for the opportunity to review the manuscript “Effectiveness of Psychological Treatments for Borderline Personality Disorder and Predictors of Treatment Outcomes: A Multivariate Multilevel Meta-Analysis of Data from all Design Types.” This meta-analysis is unusual in a number of ways.  As the authors describe on p. 2 “Instead of meta-analyzing the effect sizes representing the difference between two study arms in an RCT, we therefore meta-analyzed the effect sizes representing the change on BPD indices within study arms.”  This “allows the inclusion of the large number of uncontrolled studies that characterize the BPD treatment literature” and so they include RCTs as well as open trials, case series or cohort studies.  Despite the effects being within-treatment, they note this “still allows comparisons between treatment approaches and other predictors across studies. In fact, even allows for comparisons between treatments and certain predictors that cannot be made based on traditional meta-analyses.”  In order to make these between-treatment comparisons, they calculate a grand mean of outcomes across all interventions, and then run a series of contrasts of a given treatment’s mean against the grand mean.  Using this approach, for overall BPD symptoms they find that Schema Therapy (ST) and Reduced DBT were significantly more effective than the grand mean of all interventions (and only TAU was less effective than the grand mean).  In other contrasts, only ST and MBT were significantly greater than the grand mean on suicidality, only Reduced DBT was significantly greater than the grand mean on anger and affect instability.  No treatment was significantly greater than the grand mean on a number of other outcomes (abandonment, interpersonal, identity, impulsivity, emptiness, dissociation; though TAU was significantly lower than the grand mean for a few of those).  Based on these findings they conclude that “Schema Therapy, Mentalization-Based Treatment and reduced Dialectical Behavior Therapy were superior.” 

What’s notable about this conclusion is that it stands in stark contrast to every other meta-analysis on BPD treatments, which have repeatedly found that while the empirically supported treatments (ESTs) demonstrate greater effects than TAU conditions, no bonafide treatment has been shown to be superior to another (see Cristea et al. (2017). Efficacy of psychotherapies for borderline personality disorder: a systematic review and meta-analysis. JAMA Psychiatry, 74(4), 319-328).  This includes a recent meta-analysis from many of the authors of the present work, who very cautiously noted some positive effects in specific treatments but nonetheless noted that “specialized psychotherapies compared to protocolized psychological treatment show no significant differences on almost any outcome, although the quality of the evidence was mostly low to very low due to a significant amount of heterogeneity.” (See p. 958 in Oud, M., Arntz, A., Hermens, M. L., Verhoef, R., & Kendall, T. (2018). Specialized psychotherapies for adults with borderline personality disorder: a systematic review and meta-analysis. Australian & New Zealand Journal of Psychiatry, 52(10), 949-961.)  Therefore, the assertion of any EST for BPD as “superior” to another is an outlier that warrants scrutiny.  

Further, the assertion that Reduced DBT – and not the full course of DBT – is “superior” is also an outlier from a number of recent high-quality studies of this question.  Shelly McMain has recently given conference presentation on the results of a trail of Reduced DBT (McMain et al. (2018). The effectiveness of 6 versus 12-months of dialectical behaviour therapy for borderline personality disorder: the feasibility of a shorter treatment and evaluating responses (FASTER) trial protocol. BMC psychiatry, 18(1), 1-16.) which has found no difference between 6 vs 12 months of DBT.  This is consistent with Linehan’s component analysis of DBT (Linehan et al. (2015). Dialectical behavior therapy for high suicide risk in individuals with borderline personality disorder: a randomized clinical trial and component analysis. JAMA Psychiatry, 72(5), 475-482.) which found few differences in DBT reduced to skills components vs a full course of DBT.  Again, the assertion that Reduced BPD is “superior” is counterintuitive and warrants scrutiny.  

Looking at analyses, a strong argument could be made that these reported differences are an artifact of biases introduced by their meta-analytic approach, rather than true differences in the efficacies of the treatments.  Whereas traditional meta-analysis directly compares interventions within randomized controlled trials using standard effects, theirs borrows a bit from a network meta-analytic approach in that it makes indirect comparisons between treatments.  However, rather than use a network meta-analytic approach to compare indirect effects (which perhaps is challenged by the choice to include uncontrolled studies), they instead compare each treatment to a grand mean of all treatments (which is itself very unusual).  Still, like a network meta-analysis, their analytic approach necessitates the inclusion of treatment studies that are highly similar to one another in terms of study/patient characteristics, i.e., transitivity – you’d want to include as homogeneous a set of studies using patients with as few differences as possible.  This study does the opposite - by also including uncontrolled studies (open trials, case series, cohort studies) the likelihood of intransitivity is maximized.  The fact that ST was found to have such large effects, and that those effects were overwhelmingly based on small uncontrolled ST studies, is likely not a coincidence. 

Remember that effect sizes are based on the standard deviation (SD) of the sample – in a RCT the severity/range of pathology should not differ between treatment conditions within the trial, but this may differ widely across trials. Whereas in traditional meta-analysis the bias is pooled across all studies, their current approach may favor treatments with a narrow SD of pathology.  The counterintuitive finding of Reduced DBT’s larger effect size may become more understandable when compared to, for example, the complex severity observed in Linehan’s original RCTs, possibly diminishing Full DBT’s effect size even if the treatments did not observe less change – this is just a hunch, but the point is that there were likely differences in the Reduced DBT trials themselves (not their magnitude of change) that favored its effect size. 

Lastly, the assertion that “Schema Therapy, Mentalization-Based Treatment and reduced Dialectical Behavior Therapy were superior” is not consistent with the confidence intervals (CIs) that they report here (e.g., to stick with the Reduced vs Full DBT example, yes for those analyses the CI for Reduced DBT does not overlap with the CI for TAU, but the CIs for Reduced and Full DBT largely overlap with each other), so the assertion of “superiority” is inaccurate.

In summary, Oud et al 2018 meta-analysis by a number of the authors is quite solid; rather than address limitations of their prior research, this work introduces major limitations and abandons the caution of their prior work.  I would strongly encourage the authors to consider more traditional meta-analytic methods with an eye towards identifying specific gaps in the many high-quality meta-analyses already completed on this topic – otherwise there is not a clear rationale for this study. 

Reviewer 3 Report

Review Effectiveness of psychological treatments for BPD (Rameckers et al.)

The present manuscript is a comprehensive and thoughtful review of the current literature on the treatment of adult BPD.  The review deals with new and interesting treatment studies for BPD and for the first time tested the influence of several predictors like setting, age, and outcome. However, there are several areas in this manuscript that need attention and revision. My main concern is that the review contains too many technical and procedural details in some places (methods section, discussion) and not enough explanations for the findings in others. For example, the discussion section would benefit from a profound elaboration of the main findings in the context of the current literature to help readers understand the theoretical framework for the presented results. The parts that need most revisions are:

  1. Abstract: What is meant by “mean age was related to treatment outcome”? What kind of outcome? What is meant by “outcome type”? What kind of design type are you referring to?
  2. Line 45: I think this statement cannot be generalized to all BPD populations. I recommend to rephrase it – in some cases a combination of psychotherapeutic and pharmacological treatment might be useful; furthermore, I would not use this as an argument for the present meta-analysis but rather outline what is the new and unique aspect of this review compared to existing reviews in that field
  3. Line 113-124: delete ;
  4. Line 139: I would add the exclusion of adolescent BPD in the abstract
  5. Line 139: I don’t think that survive is the correct term here
  6. Line 202: Do you have any reasons to take the first one?
  7. Flowchart: What is meant by mixed samples?
  8. Line 317: Again, what is meant by that sentence?
  9. Line 466: What does the word in brackets mean?
  10. Line 497: There might be other explanations for why certain BPD domains showed less improvements (i.e. the focus of specialized treatments? Did they neglect certain problem domains of BPD?)
  11. Line 499: Are there any reasons why different treatments showed different effects on certain BPD criteria?
  12. Line 606: A further limitation of the present review is that it only focuses on adults and not on adolescents.

Round 2

Reviewer 2 Report

There are many positive developments in the revision of the manuscript “Effectiveness of Psychological Treatments for Borderline Personality Disorder and Predictors of Treatment Outcomes: A Multivariate Multilevel Meta-Analysis of Data from all Design Types.”

Most notably, I appreciate that the authors have removed the “superior/inferior” language that implied differences among the bonafide treatments. 

However, they continue to use language that implies a rank order among the bonafide treatments; on p. 11 they say “Together, these findings suggest that mainly the specialized treatments, i.e., ST, MBT and reduced DBT, appear most effective in the treatment of BPD.” On p. 19 they go on to note that “some, but not all Big-4 treatments (i.e., DBT and TFP) were related to larger treatment outcomes compared to the average of all treatments. ST, MBT and reduced DBT emerged as the treatments with the highest effect size.” 

Again, all of the bonafide treatments listed here have substantially overlapping confidence intervals, so any attempt to be sorting them in terms of “most effective” is not supported by their own data.  I understand that elsewhere they are being cautious to not directly compare these treatments to each other, but rather to compare them to the grand mean among all therapies (bonafide + TAU); however, in my view using the latter to imply a “most effective” ranking among the specialized treatments is problematic.  The lack of differences among the bonafide treatments (in the deviation contrasts in Table 2) is a direct test of their relative effectiveness (or lack thereof); the fact that a few bonafide treatments are just over a threshold of greater effectiveness than a heterogeneous mix of manualized and non-manualized treatments, while a few bonafide treatments are just under that threshold is a much less direct test of relative effectiveness.

On the other side, I think they could clarify the discussion of CBTE – on p. 15 they say “TAU and to a lesser degree CTBE, were robustly related to weaker improvements in all BPD outcome domains” – while the TAU finding does seem to be robust, CBTE was only found to be different in the analyses with outliers removed.  Since the analyses with outlier removed only appear as Supplemental Tables, the reader must search to find significantly weaker effects for CBTE and thus they do not impress as particularly robust.   

I had also raised concerns that reported differences might be an artifact of biases introduced by their meta-analytic approach.  I appreciate that the authors examined the influence of predictors such as study quality and trial type to allay such concerns.  However, on p. 17 they get closer to putting their finger on the main concern I was raising, when they note “There were, however, differences in the number of studies that were available for each of these treatments. While many DBT trials were included, the number of ST, MBT and reduced DBT studies was smaller (between 9 and 11 studies). Therefore, it is possible that the meta-analytic findings of the specialized treatments with fewer studies were more prone to sources of bias.”   This is correct, and also why there should be significant caution in asserting that those exact same treatments “i.e., ST, MBT and reduced DBT, appear most effective in the treatment of BPD.” 

Elsewhere on p. 16 they conclude that “Although the present study used a different approach compared to more traditional meta-analyses, our findings are consistent with earlier studies and with the most recent meta-analysis in this areas. In most cases, these previous studies have not found any differences between the specialized treatments and other protocolized treatments, but only with TAU.”  Yes, I agree that their data supports this conclusion, rather than those quoted above. As Luborsky would note, the dodo bird verdict seems to still be alive and well in this data.

Reviewer 3 Report

no comments

Author Response

As there are no comments by this reviewer, we have no further response to add.